

# Effects of cost surface uncertainty on current density estimates from circuit theory

Jeff Bowman[1,2], Elizabeth Adey[2], Siow Y.J Angoh[2], Jennifer E. Baici[2], Michael G.C Brown[2], Chad Cordes[1], Arthur E. Dupuis[2], Sasha L. Newar[2], Laura M. Scott[2] and Kirsten Solmundson[2]

[1] Ontario Ministry of Natural Resources and Forestry, Peterborough, Canada
[2] Trent University, Peterborough, Canada

Corresponding author
Jeff Bowman,
jeff.bowman@ontario.ca

## ABSTRACT

**Background:** Conservation practitioners are often interested in developing land use plans that increase landscape connectivity, which is defined as the degree to which the landscape facilitates or impedes movement among resource patches. Landscape connectivity is often estimated with a cost surface that indicates the varying costs experienced by an organism in moving across a landscape. True, or absolute costs are rarely known however, and therefore assigning costs to different landscape elements is often a challenge in creating cost surface maps. As such, we consider it important to understand the sensitivity of connectivity estimates to uncertainty in cost estimates.

**Methods:** We used simulated landscapes to test the sensitivity of current density estimates from circuit theory to varying relative cost values, fragmentation, and number of cost classes (i.e., thematic resolution). Current density is proportional to the probability of use during a random walk. Using Circuitscape software, we simulated electrical current between pairs of nodes to create current density maps. We then measured the correlation of the current density values across scenarios.

**Results:** In general, we found that cost values were highly correlated across scenarios with different cost weights (mean correlation ranged from 0.87 to 0.92). Changing the spatial configuration of landscape elements by varying the degree of fragmentation reduced correlation in current density across maps. We also found that correlations were more variable when the range of cost values in a map was high.

**Discussion:** The low sensitivity of current density estimates to relative cost weights suggests that the measure may be reliable for land use applications even when there is uncertainty about absolute cost values, provided that the user has the costs correctly ranked. This finding should facilitate the use of cost surfaces by conservation practitioners interested in estimating connectivity and planning linkages and corridors.

## INTRODUCTION

Habitat loss and fragmentation can lead to increased extinction risk of populations by limiting landscape connectivity, which is defined as the degree to which landscape structure facilitates or impedes movement among resource patches (*Taylor et al., 1993*). Exploring the impacts of changes in landscape connectivity is a significant conservation research focus, because enhancements to connectivity have the potential to help mitigate extinction risk (*Crooks et al., 2017*). For example, the creation of a connected network of protected areas in India resulted in a 2,385 km$^2$ increase of critical habitat for tigers (*Panthera tigris*) (*Gubbi et al., 2015*).

In recent years, evaluating landscape connectivity has often been done through the use of a cost surface, which is a 2-dimensional depiction of a landscape where the landscape elements are assigned relative costs that are meant to represent what organisms experience as they move through the landscape (*Koen, Bowman & Walpole, 2012*). Typically, higher costs indicate higher risk, and therefore lower probability of moving successfully through the element. Cost surfaces are generally depicted as a raster of pixels, where each pixel is assigned a cost. Different movement models, such as circuit theory or least cost path analysis (*Adriaensen et al., 2003*; *McRae, 2006*), are then used to estimate connectivity between locations based on the raster of cost values. Cost assignment is often driven by species-specific considerations, because connectivity is generally a species-specific construct (*Taylor et al., 1993*).

A challenge in developing accurate cost surfaces, and therefore in modeling connectivity, is uncertainty in assigning costs to different landscape elements. True costs are rarely known, and therefore a variety of methods are used to estimate them, including expert opinion, empirical movement data, and gene flow (*Spear et al., 2010*; *Koen, Bowman & Walpole, 2012*). The accuracy of cost values may be evaluated through comparison with field data (*Cushman et al., 2006*; *Lee-Yaw et al., 2009*), but even in such cases, it is often the relative, rather than absolute costs that are learned.

Given uncertainty in cost values, it seems important to understand the sensitivity of different connectivity estimates to variations in the estimated costs (*Adriaensen et al., 2003*). For example, *Rayfield, Fortin & Fall (2010)* found that the sensitivity of placing least-cost routes is positively related to the amount of habitat fragmentation and inversely related to the hospitality of the matrix. *Beier, Majka & Newell (2009)* demonstrated that if a model has the correct rank order of costs, the placement of a least-cost path is robust to uncertainty. They recommended that users undertake sensitivity analyses when developing cost surfaces. *Koen, Bowman & Walpole (2012)* showed that accumulated cost of least-cost paths and effective resistance estimates from circuit theory (*McRae et al., 2008*) are more sensitive to changes in cost surface than least-cost paths measured as Euclidean distance. Most recently, *Zeller et al. (2017)* showed that current density measures from circuit theory are sensitive to landscape definition, including spatial grain, thematic resolution, and the number of geospatial layers. Varying these aspects of landscape definition can alter the spatial pattern of current density.

Circuit theory is used to measure landscape connectivity via the analogous relationship between electricity traveling through a circuit and a random walk (*Doyle & Snell, 1984*). The circuit theory model has increased in popularity in recent years, following the development of the Circuitscape software package (*McRae et al., 2008*). Circuit theory differs from least-cost path analysis in that multiple paths from a destination to a source are modeled rather than the single, least-cost route, and the probability of moving through any pixel on a map is estimated. This method may be preferable in cases where the assumption of traveling on the least-cost path is difficult to meet (*Adriaensen et al., 2003*), such as animals requiring knowledge of the optimal route (*Marrotte et al., 2017*). For example, gene flow is often modeled using circuit theory (*McRae & Beier, 2007*).

Circuit theory analyses provide two outputs of particular interest for connectivity studies. The first output is effective resistance, which is a pairwise measure of cost distance between nodes and is measured as the voltage induced by passing a 1 amp current between nodes (*Doyle & Snell, 1984*; *McRae et al., 2008*). This is equivalent to commute time, or the expected time needed to get from source to destination and back again. Effective resistance is often used to model genetic connectivity between individuals or populations (*McRae & Beier, 2007*; *Lee-Yaw et al., 2009*; *Spear et al., 2010*), but is not a mappable quantity. The second output, current density, is a localized measure of movement probability at a spatially-referenced point that can be interpreted as the probability of a location being used during a random walk between a source and destination node. Current density can be mapped and is thus particularly useful for applications where mapped routes are of interest, such as in natural heritage planning (*Koen et al., 2014*), modeling animal movement routes (*Walpole et al., 2012*; *Marrotte, Bowman & Wilson, 2020*), or forecasting migration arising from shifts in climate (*Lawler et al., 2013*). Recently, *Marrotte et al. (2017)* used current density to estimate at-site genetic connectivity of sampled nodes.

Multiple sources of uncertainty exist when developing a cost surface to estimate current density, including uncertainty about relative cost weights of different landscape elements, and the thematic resolution (i.e., number of classes) of the cost surface map. Uncertainty in assigning costs has been identified as a difficult and often subjective problem in modeling connectivity (*Peterman, 2018*; *Grafius et al., 2017*; *Etherington, 2016*). Variability in the relative cost weights assigned to particular landscape elements might affect current density, and since absolute costs are often unknown, the sensitivity of current density estimates to this variation is an important consideration. If current density estimates are highly sensitive to such variation, then a priority must be placed by users on accurate estimation of relative or absolute costs. On the other hand, if current density is insensitive to variation in relative cost weights, then current estimates should be robust to uncertainty. *Beier, Majka & Newell (2009)* showed that, provided rank order is correct, least-cost route placement will be correct, but we are unaware of similar analysis evaluating current density. A contrast to the rank order of costs being correct occurs where the spatial arrangement of costs is varied, while other aspects of the cost surface are maintained. Varying spatial arrangement of costs can influence the placement of a least-cost route (*Rayfield, Fortin & Fall, 2010*), and we anticipate a similar effect for circuit theory analysis. There might also be an influence of the range of relative cost weights on current density

**Table 1 Cost scenarios used to compare across simulated and urban landscapes with three cost categories.**

| Scenario | Low | Medium | High | Range |
|---|---|---|---|---|
| C1 | 0.1 | 0.5 | 1 | 0.99 |
| C2 | 1 | 1.5 | 2.25 | 1.25 |
| C3 | 1 | 1.5 | 150 | 149 |
| C4 | 1 | 2 | 3 | 2 |
| C5 | 1 | 2 | 200 | 199 |
| C6 | 1 | 5 | 7.5 | 6.5 |
| C7 | 1 | 5 | 500 | 499 |
| C8 | 1 | 100 | 150 | 149 |
| C9 | 1 | 100 | 10,000 | 9,999 |
| C10 | 10 | 100 | 1,000 | 990 |

estimates. *Beier, Majka & Newell (2009)* referred to this concept as dispersed versus compressed cost values, finding that least-cost path analysis was robust to this type of uncertainty. Finally, landscape definition, including spatial and thematic resolution, and the number of geospatial data layers employed, can affect estimates of current density. *Zeller et al. (2017)* found that varying these characteristics of landscape models produced different estimates of where the highest current density pixels occurred.

We evaluated the effects of varying relative cost weights, the range of cost weights, the spatial arrangement of cost classes, and thematic resolution on current density estimates from circuit theory. We predicted that, as with least-cost paths (*Beier, Majka & Newell, 2009*), current density would not be sensitive to relative cost weights or the range of cost weights provided that the rank order of the costs was not changed. We further predicted that changing the spatial arrangement of relative cost weights would affect current density. Finally, we predicted that current density would not be sensitive to thematic resolution if the rank order of costs was maintained across themes.

## MATERIALS AND METHODS

We generated cost surface maps to test the sensitivity of current density estimates to changing cost values, range of cost values, varying spatial arrangement, and the number of cost classes (or thematic resolution). We started with eight cost surfaces using values from *Rayfield, Fortin & Fall (2010)*, who selected cost scenarios that represented the range of values used in connectivity studies. We generated eight cost surface scenarios, each with three levels (low, medium and high cost; Table 1). We also generated two additional scenarios using alternative low-cost values (0.1, 10) that were not used by *Rayfield, Fortin & Fall (2010)*. For all simulated maps, we assigned 25% of cells low cost values, 25% medium cost values and 50% high cost values.

We first used these 10 cost scenarios to evaluate the effects of relative cost weight and the range of cost weights on current density estimates. We did this at three different levels of fragmentation, high, medium and low (Fig. 1). In the highly fragmented treatment,

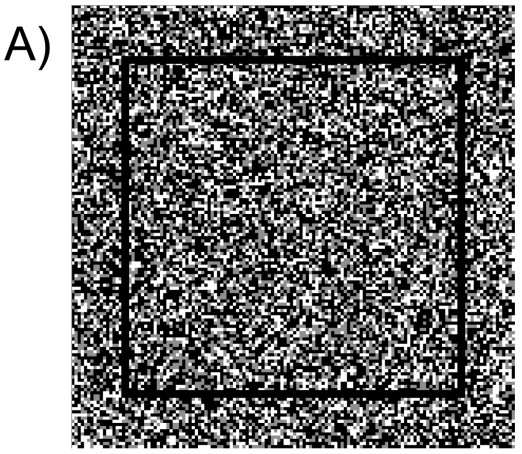

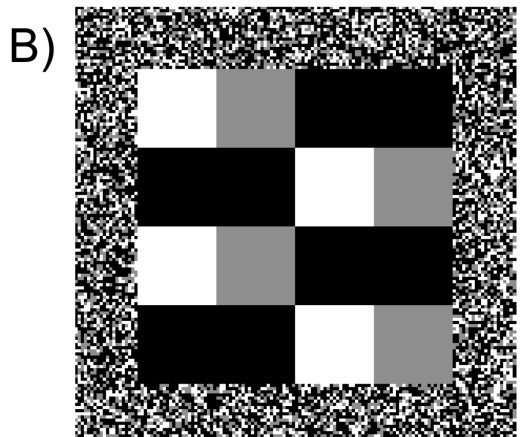

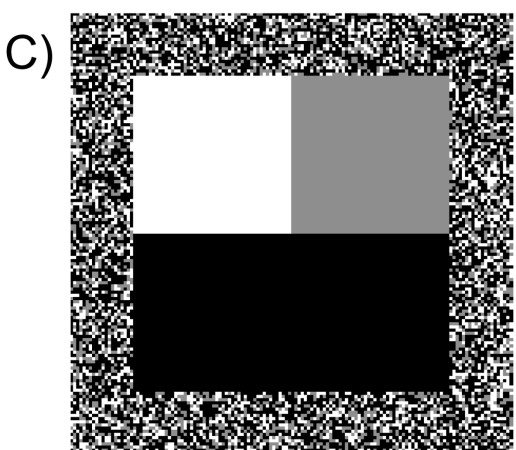

**Figure 1 Simulated cost surfaces with three levels of fragmentation.** Cost surfaces with $100 \times 100$ pixels and a 20-pixel buffer around the perimeter ($140 \times 140$ cells total). All maps had a total ratio of 25% low cost (white), 25% medium cost (grey) and 50% high cost (black). Treatments included (A) High fragmentation (randomized), (B) Medium fragmentation and (C) Low fragmentation. Each cell was assigned a weight corresponding to its cost. For each level of fragmentation, we generated 10 maps with varying cost weights.
each pixel was randomly assigned to one of the three cost values. We produced 10 different high fragmentation maps—one for each cost scenario. Medium fragmentation maps were produced by placing 12 blocks of contiguous cost values within each map (10 maps in total), and low fragmentation maps had three contiguous blocks (again, 10 maps in total) (Fig. 1). We used Spearman rank correlations to compare current density values across all 10 cost scenarios within fragmentation levels. As a contrast to varying the relative cost weights, we used the same 10 scenarios and three fragmentation levels to explore the effects of changing spatial arrangement of cost classes on current density estimates. Changing spatial arrangement of costs is one form of rearranging cost ranks, allowing us to compare different configurations of cost surfaces with the same overall costs and proportions of classes (50% high, 25% medium and 25% low). For this analysis we used Spearman rank correlations to compare current density values between fragmentation levels.

The range of cost values on a map might affect current density estimates by compressing or dispersing the costs (Beier, Majka & Newell, 2009). If this is the case, the effects might be caused by the high cost classes, since low costs are bounded by 0 in our design, and high costs have no such bound (Koen, Bowman & Walpole, 2012). To test this idea, we evaluated the effect of the absolute value of the difference in the range of costs between scenarios on the Spearman correlation between scenarios for the simulated landscapes (Table 1). We also tested for a relationship between the maximum current density from each scenario and the range of costs. We tested for this relationship only within each fragmentation level (high, medium and low).

We also sought to investigate our questions related to the effect of relative cost weights and the range of cost weights on a real-world landscape, so we tested the effects of varying cost weights on current density estimates in a 107-km$^2$ urban study area encompassing Oakville, Ontario. The area was a highly fragmented urban landscape, consisting of 10% low, 14% medium and 76% high cost values. Designation of cost values for the urban study area followed the methodology of Bowman & Cordes (2015) using the Southern Ontario Land Resource Information System (SOLRIS) database (Ontario Ministry of Natural Resources, 2008). We varied costs in the urban landscape using the same 10 cost structures employed in the simulated landscapes, producing 10 different maps. The resolution of the map was 15 m.

We created an additional 20 simulated maps to test the effect of thematic resolution (3, 6, 12 cost classes per map) on current density (Fig. 2). We started with the low fragmentation scenario with the three thematic categories that we had already created. To generate six cost categories, we divided the original three cost values from the low fragmentation map into portions reflecting 45% and 55% of the costs per theme. These six subcategories were then divided by 48% and 52% to further stratify the cost surface map into 12 cost categories (Table 2). These proportions were selected to maintain the rank order of the original three class map, while also varying thematic resolution. We then compared outputs across these three different scenarios (three, six, or 12 categories).

All simulated maps were 140 × 140 cells, consisting of 100 × 100 cells with a 20-cell buffer, to accommodate edge effects (Koen et al., 2010) on the nodes. Koen et al. (2014) showed that a buffer equal to 20% of the width of the study area would largely remove the

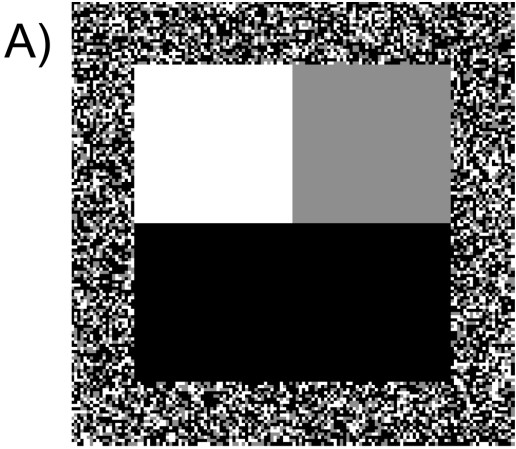

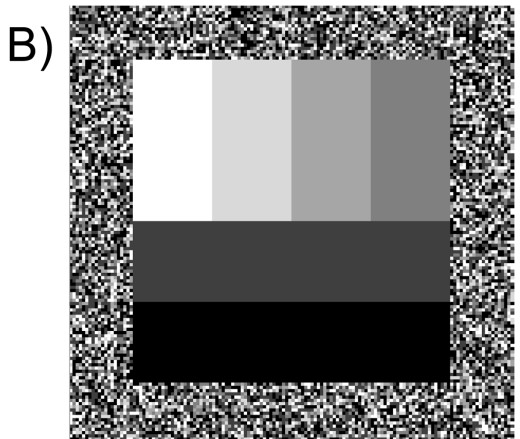

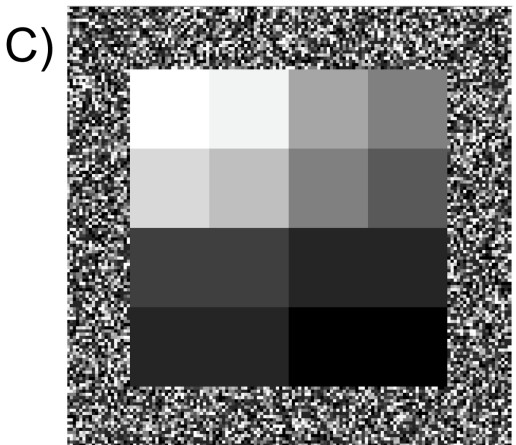

**Figure 2 Simulated cost surfaces with varying thematic resolution (three, six and 12 categories).** Cost surfaces with 100 × 100 pixels and a 20-pixel buffer around the perimeter (140 × 140 cells total). All maps had a total ratio of 25% low cost (white), 25% medium cost (grey) and 50% high cost (black). Treatments included three levels of thematic resolution (A) three categories, (B) six categories and (C) 12 categories. Each cell was assigned a weight corresponding to its cost. For each level of thematic resolution, we generated 10 maps with varying cost weights.

Table 2 Cost scenarios used to compare across six and 12 category thematic resolutions while maintain rank order of costs in comparison to a three category cost surface.

| | Six categories | | | | | | Twelve categories | | | | | | | | | | | | |
|---|---|---|---|---|---|---|---|---|---|---|---|---|---|---|---|---|---|---|
| | Lower | | Medium | | Higher | | Lower | | | | Medium | | | | Higher | | | |
| C1 | 0.045 | 0.055 | 0.225 | 0.275 | 0.45 | 0.55 | 0.0216 | 0.0234 | 0.0264 | 0.0286 | 0.108 | 0.117 | 0.132 | 0.143 | 0.216 | 0.234 | 0.264 | 0.286 |
| C2 | 0.45 | 0.55 | 0.675 | 0.825 | 1.0125 | 1.2375 | 0.216 | 0.234 | 0.264 | 0.286 | 0.324 | 0.351 | 0.396 | 0.429 | 0.486 | 0.5265 | 0.594 | 0.6435 |
| C3 | 0.45 | 0.55 | 0.675 | 0.825 | 67.5 | 82.5 | 0.216 | 0.234 | 0.264 | 0.286 | 0.324 | 0.351 | 0.396 | 0.429 | 32.4 | 35.1 | 39.6 | 42.9 |
| C4 | 0.45 | 0.55 | 0.9 | 1.1 | 1.35 | 1.65 | 0.216 | 0.234 | 0.264 | 0.286 | 0.432 | 0.468 | 0.528 | 0.572 | 0.648 | 0.702 | 0.792 | 0.858 |
| C5 | 0.45 | 0.55 | 0.9 | 1.1 | 90 | 110 | 0.216 | 0.234 | 0.264 | 0.286 | 0.432 | 0.468 | 0.528 | 0.572 | 43.2 | 46.8 | 52.8 | 57.2 |
| C6 | 0.45 | 0.55 | 2.25 | 2.75 | 3.375 | 4.125 | 0.216 | 0.234 | 0.264 | 0.286 | 1.08 | 1.17 | 1.32 | 1.43 | 1.62 | 1.755 | 1.98 | 2.145 |
| C7 | 0.45 | 0.55 | 2.25 | 2.75 | 225 | 275 | 0.216 | 0.234 | 0.264 | 0.286 | 1.08 | 1.17 | 1.32 | 1.43 | 108 | 117 | 132 | 143 |
| C8 | 0.45 | 0.55 | 45 | 55 | 67.5 | 82.5 | 0.216 | 0.234 | 0.264 | 0.286 | 21.6 | 23.4 | 26.4 | 28.6 | 32.4 | 35.1 | 39.6 | 42.9 |
| C9 | 0.45 | 0.55 | 45 | 55 | 4500 | 5500 | 0.216 | 0.234 | 0.264 | 0.286 | 21.6 | 23.4 | 26.4 | 28.6 | 2160 | 2340 | 2640 | 2860 |
| C10 | 4.5 | 5.5 | 45 | 55 | 450 | 550 | 2.16 | 2.34 | 2.64 | 2.86 | 21.6 | 23.4 | 26.4 | 28.6 | 216 | 234 | 264 | 286 |

effect of node bias in circuit analyses. The urban study area was comprised of 476,527 cells with an additional 726,423 cells to accommodate the required buffer equal to 20% the width of the area.

We used Circuitscape v4.0 (*McRae, Shah & Mohapatra, 2013*) to construct a current density map by simulating electrical current flow iterated between all pairs of nodes on a rasterized surface. We connected the eight neighboring cells as an average cost using the pairwise mode and extracted the cumulative current density map. The current density within a pixel, as a result of simulated electrical current flowing across the surface is proportional to the probability of animal movement (*McRae et al., 2008*). To model omnidirectional movement in the simulated landscapes, we distributed 20 nodes in equal numbers among four sides of the surface in the 20-cell buffer, using the same stratified placement of nodes for all scenarios. Due to the increased size and more complex shape of the Oakville study area, 100 nodes were distributed in a stratified placement around the buffered cost surface. See *Marrotte et al. (2017)*, *Bowman & Cordes (2015)* and *Koen et al. (2014)* for additional descriptions of the point-based omnidirectional methodology used.

We note that circuit theory users will most often be interested in relative, rather than absolute differences in current density. In fact, current density outputs are often standardized for ease of comparison across landscapes (e.g., *Bowman & Cordes, 2015*). Therefore, our method of comparing current density outputs was aimed at evaluating relative differences in the distribution of current across maps. In ArcMap v10.5 (*ESRI, 2016*), we produced a polygon over the generated study area and randomly selected 1,000 data points within this polygon. We then measured the correlation of the current density values of these randomly selected data points across all maps by creating a Spearman rank correlation coefficient matrix in R v3.4.1 (*R Core Team, 2014*). We compared the mean correlation coefficients and the range of correlations across scenarios of interest.

## RESULTS

In our simulated landscapes, the three fragmentation levels had equivalent costs for any given cost scenario, varying only in the spatial arrangement of these costs. Mean (range) current density for each of the three levels was 1.36 amperes for high fragmentation (1.25–1.59; $n = 10$ scenarios), 1.37 amperes for medium fragmentation (1.27–1.49), and 1.0 ampere for low fragmentation (0.54–1.25). The six and 12 class surfaces had mean values of 1.20 amperes (1.11–1.26) and 1.21 amperes (1.13–1.26), respectively. The urban landscape had a mean (range) current density of 3.95 amperes (3.78–4.31; $n = 10$ scenarios).

We found that current density values were highly correlated within fragmentation levels across cost scenarios (Table 1; Fig. 3). High, medium and low fragmentation levels had mean Spearman correlation coefficients (range) of 0.87 (0.65–1.00), 0.90 (0.80–1.00) and 0.90 (0.74–1.00), respectively ($n = 45$ comparisons for each analysis). Similarly, the urban study area (Fig. 4) had a mean Spearman correlation coefficient (range) of 0.85 (0.62–1.00) ($n = 45$ comparisons).

When the rank order of costs was not maintained, by changing the spatial arrangement of the costs, correlations were reduced. The correlation coefficients for comparisons between fragmentation levels (range) were −0.03 (−0.17 to 0.06) for the comparison of low and medium fragmentation, −0.07 (−0.10 to −0.04) for medium and high, and −0.02 (−0.12 to 0.10) for low and high levels of fragmentation ($n = 100$ for each comparison) (Fig. 3).

In simulated landscapes, the correlation between current density values was affected by the range of cost values (Fig. 5). In general, correlations were more variable when the range of cost values was broader, although, correlations always exceeded $R_S = 0.65$. A broad range of cost values occurred when the highest cost class was especially high, and this led to the occurrence of especially high current density values (Fig. 6). Maximum current density in a simulated landscape was correlated with cost range: $R_S = 0.85$, 0.80 and 0.57 for high, medium and low fragmentation levels, respectively ($n = 10$ for each comparison). High current density was associated with especially high cost classes, leading to overall high variability in current density, reducing correlations with other cost scenarios. Maximum current density in the urban landscape was also correlated with cost range ($R_S = 0.78$, $n = 10$ scenarios; Fig. 6).

Within scenarios with six or 12 thematic classes, current density was highly correlated across cost values: $R_S$ (range) = 0.90 (0.74–1.00) and 0.92 (0.80–1.00), for six and 12 cost classes, respectively ($n = 45$ comparisons for each analysis). Comparing scenarios with different thematic resolution, the Spearman correlation coefficients (range) were 0.87 (0.69–0.97) for three and six classes, 0.89 (0.75–0.98) for six and 12, and 0.85 (0.71–0.97) for three and 12 ($n = 100$ for each analysis) (Fig. 7).

## DISCUSSION

We applied circuit theory to measure the sensitivity of current density estimates to varying cost surfaces, including the relative cost weights, the range of cost weights, the spatial arrangement of the classes throughout the landscape (fragmentation), and the number of

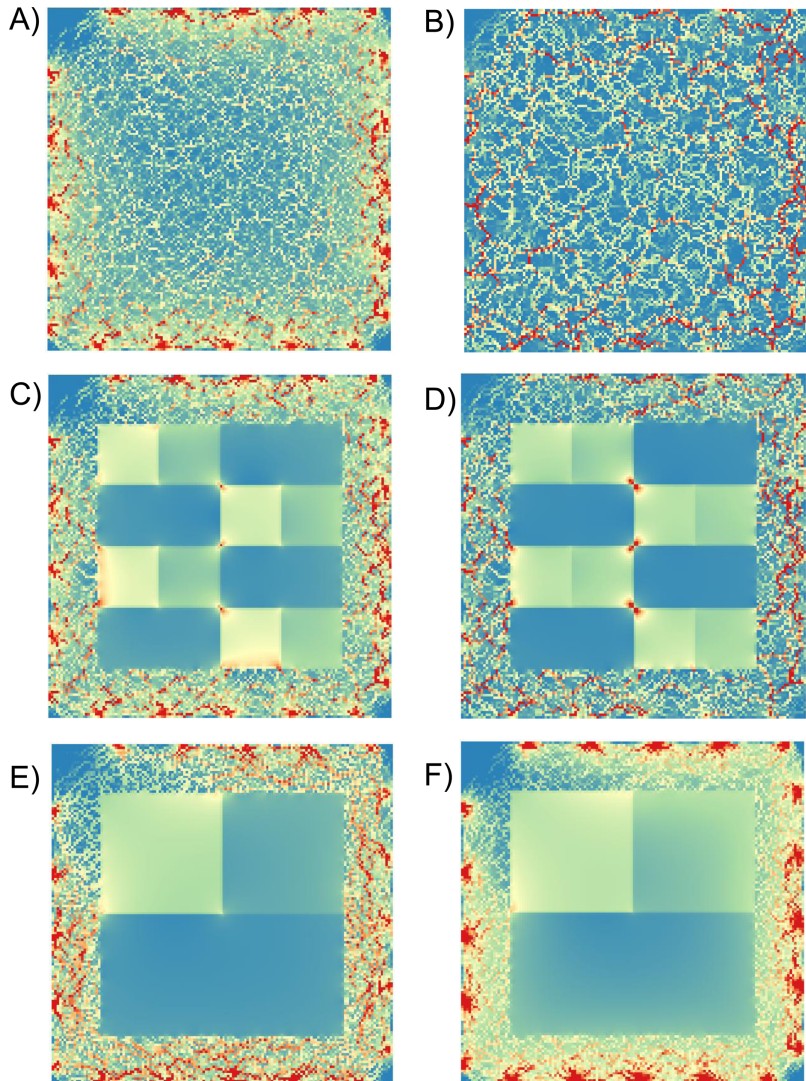

**Figure 3** **Example current density maps for cost surfaces at three levels of fragmentation.** Examples of current density maps at a resolution of 140 × 140 cells for landscapes with different fragmentation levels (high, medium and low), different cost classes, and a 20-pixel buffer. (A), (C) and (E) had a costs of 7.5 (high), 5 and 1 (low), whereas (B), (D) and (F) had costs of 1,000, 100, 10, respectively. Highest current density is indicated by red. Node locations are indicated by areas of high current density around perimeters, especially evident in (A) and (F).

classes (thematic resolution). Given that the placement of least cost routes is robust to uncertainty in relative cost weights and the range of costs (*Beier, Majka & Newell, 2009*), we suspected that varying cost weights would not substantially alter current density patterns from circuit theory when relative ranks of costs were held constant. For a given level of fragmentation, current density was highly correlated across landscape models despite differences in cost values. Thus, relative current density estimates were not sensitive to cost value. This was true both for simulated landscapes and a real-world urban landscape. The range of cost values had some effect on current density however, as there was an inverse relationship between the ranges of cost values in a landscape and the

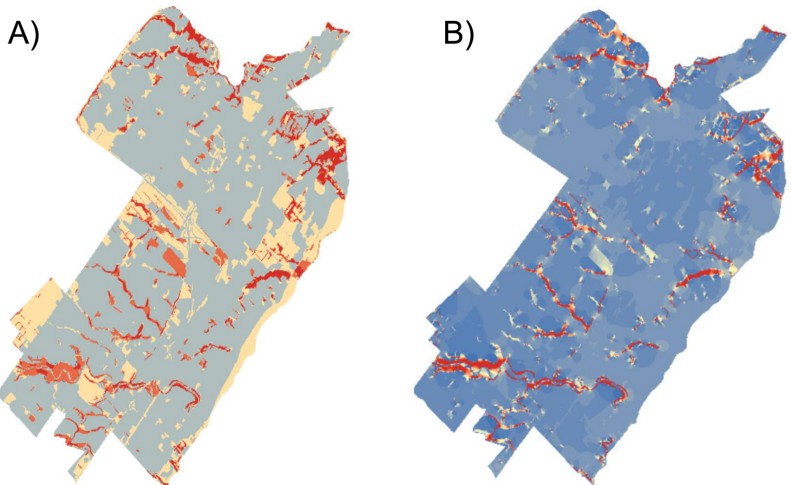

**Figure 4 Example current density maps from an urban landscape.** Two examples of current density maps for an urban study area in Oakville, Ontario: (A) had costs of 1 (low), 2 and 3 (high) and (B) had costs of 1, 100 and 150. The Spearman rank correlation was 0.68, lowest of all comparisons for this landscape (mean (range) $R_S$ = 0.85 (0.62–1.00) $n$ = 45 comparisons).

amount of correlation in current density of landscapes. We did find that changing the spatial arrangement of cost classes had a high impact on current density, as shown by the low correlation between different levels of fragmentation in our maps. Current density is sensitive to landscape configuration. However, high correlations between current density estimates arising from scenarios with varying numbers of classes suggests that thematic resolution itself has little impact on current density, provided that the rank order of the costs is maintained across themes.

Our finding of low sensitivity to varying relative cost weights in circuit theory concurs with *Beier, Majka & Newell (2009)*'s finding with least cost paths, in that provided the costs had a consistent rank order, the current density maps were quite similar. This is a useful result, insofar as true costs are often unknown by users (*Grafius et al., 2017*; *Bishop-Taylor, Tulbure & Broich, 2015*; *Yumnam et al., 2014*). Therefore, where users have insight about the relative costs of elements in the landscape, current density outputs should be reliable. This is helpful for applications where current density is employed in natural heritage planning or designing wildlife linkages (*Marrotte et al., 2017*; *Bowman & Cordes, 2015*).

The range of cost values used in a map did affect the correlation between current density values. As the range increased, current tended to be more highly pinched, whereas it spread more as the range decreased (e.g., Fig. 3). This observation is further supported by the overall increase in the variation of correlation results as the absolute cost value range increased (Fig. 5). The cost range influenced current density through the effect of high cost weights. In our simulations, cost was limited by 0 at the lower end, but was variable across scenarios at the upper end (*Koen, Bowman & Walpole, 2012*). It was surfaces with the highest cost that also had the broadest range of costs (i.e., the costs were more dispersed),

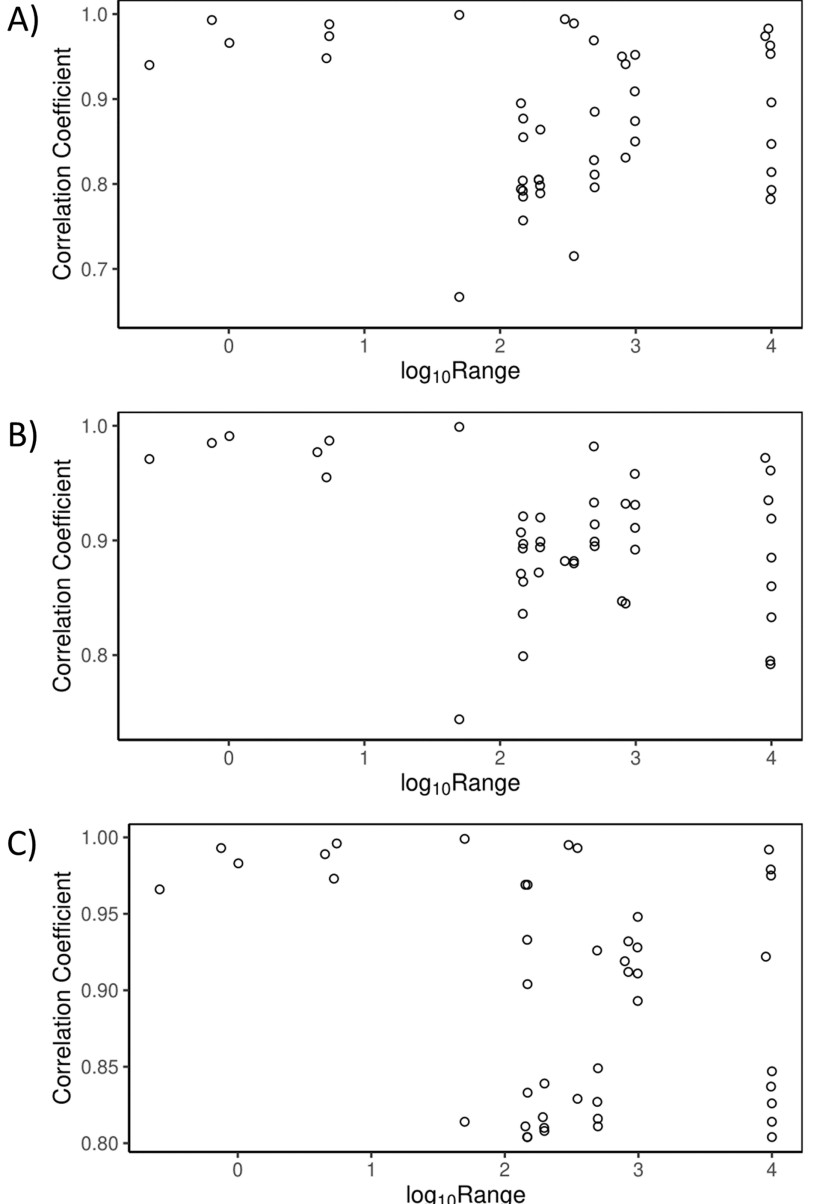

**Figure 5  Effect of the range of cost values on correlation of current density across landscapes.** Effect of the absolute value of the difference in the range of cost values in a landscape (log$_{10}$-transformed) on the Spearman correlation between current density estimates in pairs of landscapes: (A) high, (B) medium and (C) low fragmentation.                                               

and these surfaces produced higher current density values than comparable surfaces with a narrower (i.e., compressed) cost ranges (Fig. 6). We are reminded with this observation that high current density is a result of high cost weights (*Marrotte et al., 2017*). Despite variation in current density values however, the Spearman correlations between landscapes were always rather high within the fragmentation scenarios, such that there was always agreement among maps, even where the range of cost values was high. The lowest correlation between any pair of maps within fragmentation levels was $R_S = 0.65$.

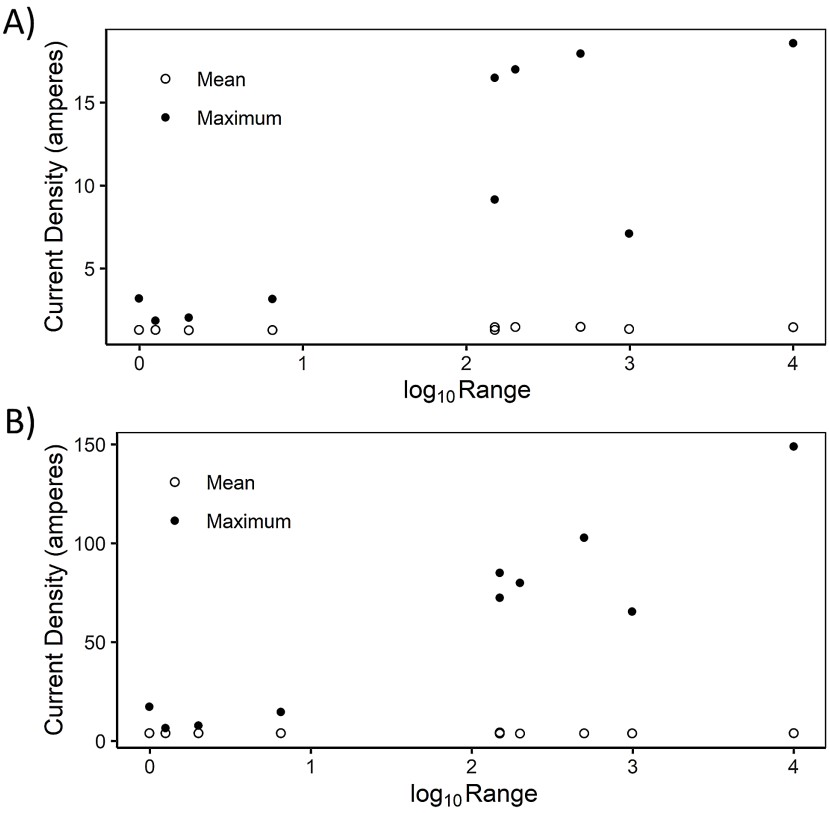

**Figure 6 Effect of cost range on current density.** Effect of the absolute value of the difference in the range of cost values in a landscape (log$_{10}$-transformed) on mean and maximum current density estimates cross 10 different cost scenarios. We have shown here (A) the medium fragmentation scenarios in a simulated landscape and (B) the real, urban landscape.

Our findings also concur with those of *Rayfield, Fortin & Fall (2010)*, who found that the location of least-cost paths varied with fragmentation. In particular, they showed that routes were more sensitive to higher amounts of fragmentation, or lower quality matrix. We found that altering the configuration of the landscape, while holding overall costs constant, affected the distribution of current density. We consider that the low correlation across fragmentation scenarios was due to the changing spatial arrangement of landscape elements, which altered the location of low and high cost routes. Changing the spatial arrangement served to change the rank order of the cost categories, so this treatment served to show the effect of altering rank order.

We add to the work of *Zeller et al. (2017)*, who found that current density estimates were sensitive to landscape definition, which they varied by changing the data source, thematic resolution and spatial grain. We did not explore the effect of spatial grain in our study, but *Koen, Ellington & Bowman (2019)* have recently shown that reducing tile size can reduce accuracy of current density maps compared to untiled maps of larger extent. Our findings suggest that current density estimates are not particularly sensitive to thematic resolution, or relative cost weights, provided that the rank order of the costs does not vary. Coarsening of thematic resolution reduces detail, as similar categories are merged, but this can come with the potential benefit of increasing accuracy of ranking cost

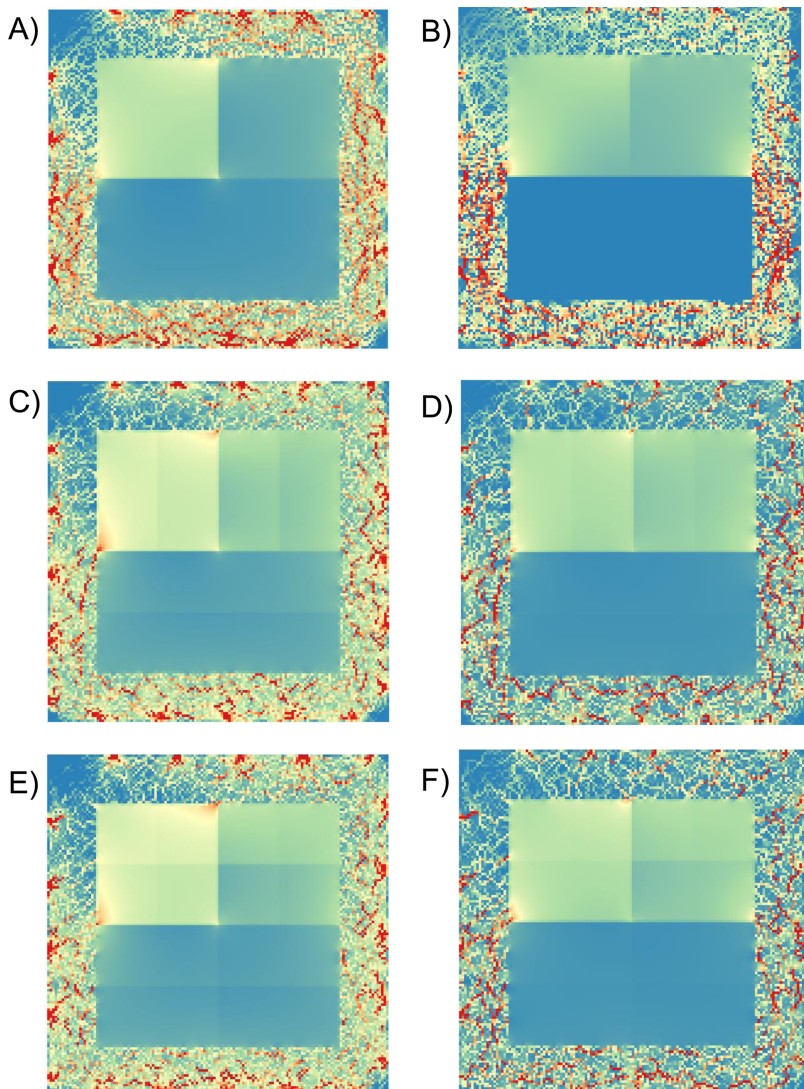

**Figure 7 Examples of current density maps with different thematic resolutions.** Examples of current density maps at a resolution of 140 × 140 cells for landscape depictions with different thematic resolutions (three (A, B), six (C, D), or 12 (E, F) categories), different costs, and a 20-pixel buffer. (A), (C) and (E) had costs of 1 (low), 5 and 7.5 (high), whereas (B), (D) and (F) had costs of 10, 100, 1,000, respectively. Highest current density is indicated by red. Node locations are indicated by areas of high current density around perimeters.             

weights of the categories. More categories likely lead to increased error in ranking costs. Therefore, we recommend that users strike a balance between adequate detail and accurately ranked costs.

Our results suggest that current density is an effective tool for conservation planning, as it is robust to uncertainty in absolute cost values and sensitive to habitat fragmentation. Fragmentation is an increasing conservation concern as the human footprint continues to expand; a recent global analysis revealed animal movements are reduced by approximately 50% in areas with a high human footprint (*Tucker et al., 2018*). Thus, understanding the movement of animals in and across complex landscapes will be imperative for effective

conservation of biodiversity in the age of the Anthropocene. Our findings also suggest that when planning landscape connectivity applications, managers and researchers should put more effort toward establishing confidence in the rank order of costs, than reducing uncertainty in relative or absolute cost weights.

## CONCLUSIONS

We tested the hypothesis that current density estimates from circuit theory are not sensitive to absolute values of underlying cost surfaces, provided that the rank order of the costs is accurate. We compared a variety of cost surfaces with different costs, where rank order of costs was held constant, and found a high correlation among resulting current density estimates. Our findings suggest that knowledge of the absolute values of cost surfaces may not be required, provided that rank orders of costs are known. This finding should facilitate the use of cost surfaces by conservation practitioners interested in estimating connectivity and planning linkages and corridors.

## ACKNOWLEDGEMENTS

We thank Dr. Erin L. Koen for providing advice on an earlier version of the manuscript, and Erica Newton for assistance with data.

### Funding

This work was supported by an NSERC CREATE Grant to Trent University, an NSERC Discovery Grant to Jeff Bowman, and by the Ontario Ministry of Natural Resources and Forestry. The funders had no role in study design, data collection and analysis, decision to publish, or preparation of the manuscript.

### Grant Disclosures

The following grant information was disclosed by the authors:
NSERC CREATE Grant to Trent University, an NSERC Discovery Grant to Jeff Bowman.
Ontario Ministry of Natural Resources and Forestry.

### Competing Interests

The authors declare that they have no competing interests.

### Author Contributions

- Jeff Bowman conceived and designed the experiments, performed the experiments, analyzed the data, prepared figures and/or tables, authored or reviewed drafts of the paper, and approved the final draft.
- Elizabeth Adey conceived and designed the experiments, performed the experiments, analyzed the data, prepared figures and/or tables, and approved the final draft.
- Siow Y.J. Angoh conceived and designed the experiments, performed the experiments, analyzed the data, prepared figures and/or tables, authored or reviewed drafts of the paper, and approved the final draft.

- Jennifer E. Baici conceived and designed the experiments, performed the experiments, analyzed the data, authored or reviewed drafts of the paper, and approved the final draft.
- Michael G.C. Brown conceived and designed the experiments, performed the experiments, analyzed the data, authored or reviewed drafts of the paper, and approved the final draft.
- Chad Cordes conceived and designed the experiments, performed the experiments, analyzed the data, prepared figures and/or tables, and approved the final draft.
- Arthur E. Dupuis conceived and designed the experiments, performed the experiments, analyzed the data, prepared figures and/or tables, authored or reviewed drafts of the paper, and approved the final draft.
- Sasha L. Newar conceived and designed the experiments, performed the experiments, analyzed the data, prepared figures and/or tables, authored or reviewed drafts of the paper, and approved the final draft.
- Laura M. Scott conceived and designed the experiments, performed the experiments, analyzed the data, authored or reviewed drafts of the paper, and approved the final draft.
- Kirsten Solmundson conceived and designed the experiments, performed the experiments, analyzed the data, prepared figures and/or tables, authored or reviewed drafts of the paper, and approved the final draft.

## Data Availability

The data is available in the Supplemental Files.

## Supplemental Information

Supplemental information for this article can be found online at http://dx.doi.org/10.7717/peerj.9617#supplemental-information.

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
