# Peer review of "Effects of cost surface uncertainty on current density estimates from circuit theory"

_PeerJ, doi:10.7717/peerj.9617_

## Round 0.1 · original submission · Major Revisions

As you can see I have now received two very detailed reviews for your paper. Reviewer 2 (Nathaniel Pope) raises several important fundamental and methodological issues that would need to be fully addressed before I can make a decision.

·

Basic reporting

a. Introduction
i. The introduction was the strongest piece of the manuscript, in my opinion. I felt like the flow from one paragraph to the next was coherent and made significant progress in preparing me for the rest of the paper. Topic sentences were generally on target and logical. The last paragraph in the introduction (115-134), however, needs to be expanded and split into at least two paragraphs (115-126, and 126-134).
ii. 115-126: In 115, you begin by suggesting that you will explain the various sources of uncertainty in developing a cost surface. Then 116 to 123 explain how variability in cost weights are one source of variability. 123-126 brings in thematic resolution and other differences in layers as sources of variability. However, as I see the paper, you are fundamentally most interested in understanding how current density is affected by four variables or factors: (1) varying cost weights, (2) thematic resolution, (3) spatial arrangement/ fragmentation, and (4) range of cost weights. You build up cost weights strongly, but I think it is important to build a stronger basis for why you are studying thematic resolution, spatial arrangement, and range of cost weights.
iii. 126-134: I appreciate that you clearly laid out which variables you are studying in sentence 126-129. I also appreciated the clear expectations for what they expected to find. With a stronger paragraph preceding this, we’ll have a better idea of why we should expect these things. The only addition I would like to see to this paragraph is a statement of expectation/hypothesis of how changing cost values will affect current density. Basically, strengthen the sentence in 128-129.
b. Structure
i. Overall, the structure of the Introduction was very good, aside from the issues I mentioned above. The structure of the Methods section worked well for me as well. However, structural issues and omissions in the Results and Discussion made it more difficult to follow. Inconsistency in the order in which you address elements of the study makes it more difficult to follow. In the Introduction, you introduce the three factors you are studying, in an order that makes sense, but the lack of parallel structure in results and discussion is a major hindrance. Please revisit each of your hypotheses in the same order in both Results and Discussion and ensure that there is sufficient detail about each.
ii. Results:
1. The fragmentation and thematic resolution paragraphs are good, but should switch places. This will maintain consistent order with the introduction.
2. I would also strongly recommend to begin the Results section with an introductory summary paragraph about how varying absolute values of cost weights affected current density in general. Certainly you discuss this in other paragraphs in the context of other variables, but I don’t see why you couldn’t discuss this in general terms. For example, across all types of scenarios, how often did changing cost weights have a significant bearing on the current density?
3. The third paragraph focuses on cost value range, but is inadequate at describing your results. Give more specifics, e.g. numbers rather than simply “In general …”
iii. Discussion:
1. If you divide up parag. 1 based on which hypothesis it focuses on, it’s nearly all fragmentation, and only one sentence is on thematic resolution. Overall, there is little on thematic resolution in the discussion. Please remedy this.
2. You would be well served to stick with the same order for your hypotheses throughout the paper: 1) cost weights, 2) thematic resolution, 3) spatial arrangement, 4) range of cost weights. Without this structure, it is hard to follow and to know what the key messages are for each one.
c. Figures
i. Figures 1-5 worked fine for me in the sense that they gave a depiction of what the landscape models were like. The only change I would like to see is to show the nodes. You describe that in the text but do not show the nodes in the figures.
ii. Figure 6 was good. Additional figures that summarize the results in more ways than simply listing mean RS values would improve the paper.
d. English
i. The English is very good overall.
ii. “large scale” (line 259) is ok, but many conservation practitioners do not come from a strong geography background, and therefore I favor fine/coarse scale instead of large/small scale.
iii. The only typo I noticed was on line 50: “costs weights.”

Experimental design

a. Original primary research
i. The research question is original.
b. Research question defined, meaningful, fills knowledge gap
i. This is valuable for conservation – circuit theory has as stronger theoretical basis than least cost path analysis, so expanding our understanding of the behavior of circuit theory in various contexts is important.
c. Rigorous investigation
i. I understand that rank order of cost weights is probably more important than the relative differences between those cost weights. The numbers that are above 50% or even close to 90% seem great, but how do they compare to other models that do not maintain rank order? I am not certain that you have used baselines/controls/null models, against which we can compare these results. How do all these heterogeneous surfaces compare to a homogeneous surface? How much worse is the RS of a homogeneous surface than a bad model that maintains rank order? What if the rank order of cost weights is NOT preserved – how different would the results be? How do we determine at what point it is relatively unimportant to develop precise cost weights? The phrase “provided that the rank order of the costs is accurate” could become more meaningful in that case.
d. Methods detailed
i. Please explain the concept in lines 195-198 in more depth.

Validity of the findings

a. Data controlled, statistics sound
i. It was light on statistics, and only included Spearman correlations. These are appropriate and are rightly the main results. It is possible that there are other statistics that could draw additional inferences from the results, but I can’t think of any others.
b. Speculation identified
i. I did not notice any speculation.
c. Conclusions well stated, linked to research question and results.
i. Conclusion was appropriate.

Additional comments

This is a valuable study and look forward to seeing it in print. Please see my comments for details on what changes I think you should make to strengthen this study.

Karl Jarvis
Southern Utah University

·

Basic reporting

See "General comments".

Experimental design

See "General comments".

Validity of the findings

See "General comments".

Additional comments

When I read the abstract, I thought that this study would be assessing the sensitivity of current density or resistance distance to "mis-assignment" of cost values across classes (in terms of rank order). For example, if expert opinion is used to parameterize the resistance surface, and the expert decides that class A has higher cost than class B when the converse is true. Or equivalently, how sensitive outputs (passage probabilities) are to a non-monotonic transformation of inputs (conductance values/class). This sort of error is probably quite common in practice, and so seems useful to qualify the conditions under which inference is dramatically impacted.

Instead, what is investigated is the sensitivity of outputs to the relative difference in cost between classes, maintaining rank order. Equivalently, the sensitivity of outputs to monotonic transformations of inputs. Sensitivity is measured as average rank correlation (Spearman) of outputs, across transformations of inputs. So, the question being asked can be roughly boiled down to: does a monotonic transformation of the inputs produce a more-or-less monotonic transformation of the outputs? The answer seems to be yes (average Spearman correlations ~0.9), but I don't find this nearly as interesting or useful a the hypothetical scenario in the first paragraph. In fact, I'm surprised the average Spearman correlation is not closer to 1. I'm curious if there are parts of the graph that on average show a greater discrepancy that could be artefacts; e.g. regions nearer to the border of the raster (I realize the authors included a spatial buffer, but the 20% rule of thumb is not a hard guideline and depends on the size of the raster).

The second part of the study assesses sensitivity of output to "fragmentation", which corresponds to spatial autocorrelation/degree of clumping in cost values. There are 10 simulated surfaces in each "fragmentation treatment". For the first part of the study (assessing changes in magnitude of cost, discussed above), the outputs were compared across different transformations ("cost scenarios") of the same simulated surface: this is valid. For this second part, outputs are compared (via rank-correlation) across different simulated surfaces. I don't see how this is meaningful-- of course outputs from completely different spatial configurations will have a correlation of 0 on average, provided you generate enough simulations. Am I missing something?

The third part consists of comparing simulations where the cost classes are subdivided to create a more "continuous" resistance surface. This seems useful, and relates to the distinction between discrete land use classification and continuous spatial covariates.

The method used to simulate landscapes doesn't seem ideal, because:
A. The high-fragmentation landscapes are essentially just white noise (every cell is assigned a different class independently). That's going to result in connectivity that's quite similar to the case where all cells have the same cost, I believe. So doesn't seem like a very useful test-case.
B. The low- and mid- fragmentation landscapes are generated by assigning contiguous blocks of cells to a class. That's OK, but how does it give insight into real landscapes?
Given that only a single real landscape is included in the study, a lot of the oomph is riding on the generalizability of these simulations.

In summary, the general theme -- sensitivity of predictions about connectivity to assigned costs -- is useful. I'm less convinced that the particular questions asked here are useful. The manuscript is well-written and the figures are fine, although six figures is a bit much given the short length of the manuscript (especially as several of them are purely illustrative).

Ln 103: I suggest you use the interpretation of resistance distance as commute time (rather than voltage): it's proportional to the expected time needed to get from point A to point B and back again.

Ln 123: "unaware of similar analysis evaluating current density" -- I find that surprising given that commute time and passage probabilities are used heavily in other fields. Did you look into the computer science/graph theory literature? Commute-time distance (or variants thereof) is ubiquitous there and heavily employed in practical applications (e.g. Google's PageRank).

Ln 125: "landscape models" -- by this do you mean generative models of landscape, e.g. simulations from random fields?

Ln 132: Define what you mean by "themes"

Ln 133-134: "changing the spatial arrangement of relative cost weights" -- not clear what you mean here. Any sort of rearrangement of the raster would alter the spatial configuration of relative cost weights, no?

Ln 139: Given that the cost value scheme is central to the analysis, and that Rayfield et al 2009 is behind a paywall, please provide a bit more description of why you chose these values as a starting point.

Ln 144-151: this seems like a really simplistic way to simulate landscapes: why didn't you simulate via a random field model, were you could control the degree of spatial autocorrelation (for example, simulate a continuous-valued Matern process and "discretize" it)? For the "high-fragmentation" treatment, assigning each pixel independently will generate a surface that looks like white noise. This will, I think, result in resistance distances that are effectively very similar to log Euclidean distance.

Ln 167: "16" -> "6"

Ln 174: nitpicking ... "simulating" will mean stochastic simulation to most people. What you are doing is solving a large, sparse linear system: it's totally deterministic.

Ln 192: "Spearman rank correlation" ... so, you're asking whether a monotonic transform of the inputs produce monotonic transforms of the outputs? I'm frankly surprised that the Spearman correlation coefficients aren't closer to 1. I wonder if they would be, if you compared passage probabilities averaged across sliding windows (e.g. larger geographic areas than just individual pixels).

Ln 201-206: this seems like a pretty cursory analysis of the simulations. Are there particular parts in the graph where the change in cost value has a disproportionately high impact on the outputted passage probabilities?

Ln 206-209: But ... for comparing across fragmentation treatments, you're calculating correlations between outputs from completely different spatial configurations, right? Why would you expect the average correlation to be non-zero?

Ln 215-217: Again, I think this could benefit from a more thorough examination: *why* is the rank correlation in passage probability sensitive to range in cost values?

Ln 231-234: in line with previous comments-- this seems self-evident. Why test it?

Ln 243-244: interesting, and I see why Pearson correlation between outputs would be sensitive to concentration of current in particular cells ... but why would this change rank correlation?

---

## Round 0.2 · accepted · Accept

Both Reviewers are happy with the revised manuscript, so I am glad to accept your manuscript. I kindly ask you to:
1) Please address the new question raised by Reviewer 2.
2) Please justify your choice of number of nodes to model omnidirectional movement (you may want to briefly summarize Marrotte et al. (2017), Bowman & Cordes (2015) and Koen et al. (2014)).

·

Basic reporting

Authors have done an excellent job of communicating their findings

Experimental design

Methods are sound, and based on the approach of an important parallel study

Validity of the findings

I agree that the interpretation fits with the results

Additional comments

I have no additional comments. The authors addressed my previous comments thoroughly.

·

Basic reporting

See 'general comments'

Experimental design

See 'general comments'

Validity of the findings

See 'general comments'

Additional comments

You've adequately addressed my concerns, except for one point I am still confused about. When you say 'the spatial arrangement of costs is varied, while other aspects of the cost surface are maintained' (line 130) and 'changing spatial arrangement of costs is one form of rearranging cost ranks' (line 167-168), do you mean that cost values were swapped across classes while maintaining the spatial arrangement of classes? (e.g. the same landscape classification map, the same cost values, but different choice of which class was low/mid/high).

When I read the first version of this manuscript, I understood this analysis as something different: comparing current density across entirely different maps (pairs from the 10 simulations within a fragmentation level) for a fixed cost-weighting scheme ... for which case I fail to see why the expected rank correlation would be different from zero, or how comparisons across entirely different maps equate to 'rearranging cost ranks'.

In the methods, please clarify which of these is the case. Also, if possible, could you please include the code used to generate the simulated landscapes as a supplement (would help allay confusion such as the above)? Otherwise the manuscript is OK for publication, in my opinion.